# The cardiac autonomic response to acute psychological stress in type 2 diabetes

Nelly Lou Monzer[1]*, Mechthild Hartmann[1], Magdalena Buckert[1], Kira Wolff[1], Peter Nawroth[2,3], Stefan Kopf[2,3], Zoltan Kender[2,3], Hans-Christoph Friederich[1], Beate Wild[1]

1 Department of General Internal Medicine and Psychosomatics, University Hospital Heidelberg, Heidelberg, Germany, 2 Department of Medicine I and Clinical Chemistry, University Hospital Heidelberg, Heidelberg, Germany, 3 German Center for Diabetes Research (DZD), Heidelberg, Germany

* nelly.monzer@med.uni-heidelberg.de

**Data Availability Statement:** Data are available here: doi 10.6084/m9.figshare.19329341 doi 10.6084/m9.figshare.19329311.

## Abstract

### Background

Impaired cardiac autonomic control is common among people with type 2 diabetes. The autonomic nervous system and its regulatory influence on the cardiovascular system also play a key role in the physiological response to psychosocial stressors. It is unclear whether the disease-related impairment of cardiac autonomic control in people with type 2 diabetes affects the stress response. The aim of this study was therefore to examine the cardiac autonomic and the psychological stress response of people with type 2 diabetes compared to healthy control participants.

### Methods

We used the trier social stress test to induce stress in n = 51 participants with type 2 diabetes and n = 47 healthy controls. We assessed heart rate (HR) and heart rate variability (HRV) using six ECG samples before, during and after the stress test. We measured participants' psychological stress response using visual analogue scales.

### Results

Longitudinal multilevel models showed an attenuated HR increase in response to the stress test combined with a slower HR recovery after the stress test, in people with type 2 diabetes. This pattern was accompanied by significantly lower low frequency HRV but no differences in high frequency HRV between the groups. Additionally, people with type 2 diabetes showed an increased level of self-reported psychological tension 45 minutes after the stress test.

### Conclusions

The impairment of the autonomic nervous system found in people with type 2 diabetes is reflected in the HR response to stress—but not in the HRV response—and partially mirrored in the psychological stress response. Our results underline the importance of considering

**Funding:** The study was funded by the Deutsche Forschungsgemeinschaft (DFG, German Research Foundation [project numbers WI 4115/5-1 and 236360313 (SFB 1118)]. BW and PN received the awards. funder website: https://www.dfg.de The funding sources had no role in the design of the study; the collection, analysis, and interpretation of data; writing the report; or the decision to submit the report for publication.

**Competing interests:** The authors have declared that no competing interests exist.

the interplay of psychosocial stress and disease-related changes in the physiological stress response system in research and treatment of type 2 diabetes.

## Introduction

Impaired cardiac autonomic control is widespread among people with type 2 diabetes, especially among those suffering from diabetes-associated complications [1, 2]. Studies on autonomic function in people with type 2 diabetes frequently report an increased resting heart rate as well as decreased heart rate variability (HRV), implying reduced autonomic modulation of the cardiovascular system [3, 4].

The autonomic nervous system and its regulatory influence on the cardiovascular system also play a key role in the physiological response to psychosocial stress. Recent research has repeatedly demonstrated the importance of psychosocial stress as a risk factor in type 2 diabetes [5–7] and stress has become a popular target for therapeutic intervention in people with type 2 diabetes [8], but physiological mechanisms are still being discussed. Investigating possible disease-related changes in the stress response system could help to further understand the relationship between psychosocial stress and type 2 diabetes.

In healthy samples, the cardiac autonomic response to stress is characterized by an increase in heart rate as well as decreases in heart rate variability caused by a combination of parasympathetic withdrawal and increased sympathetic influence [9–11]. The ability of the autonomic nervous system to react quickly and appropriately to environmental demands has been termed autonomic flexibility and has been shown to be a crucial factor in stress resilience [12, 13]. Decreased vagal tone, prolonged autonomic recovery, hyperreactivity but also blunted reactivity have been associated with an increased vulnerability to the psychological and physiological consequences of stress, including depression [14], sleep disturbances [15] and impaired glycemic control [16].

To date, only one study by Steptoe et al. [17] has investigated the autonomic stress response in people with type 2 diabetes. They report a blunted heart response to psychosocial stress in participants with type 2 diabetes. However, they excluded people with signs of autonomic neuropathy, thus excluding a significant subsample, and did not assess HRV. An altered autonomic stress response in samples with type 2 diabetes is thus likely but the evidence does not suffice to make comprehensive assumptions as to how the autonomic stress response might be affected and to what extend this is reflected in people's psychological stress response. The aim of this study is therefore to investigate the autonomic stress response as well as the psychological stress response in people with type 2 diabetes compared to healthy control participants.

## Materials and methods

Data collection took place from June 2018 to July 2019 and was done within a larger study on the stress response in people with type 2 diabetes other results of which can be found in Monzer et al. [18] and Buckert et al. ([19] currently under review). Described materials and methods therefore partially overlap. The study was approved by the ethics committee of the Medical Faculty of the University of Heidelberg (S-019(2017)).

### Participants

People with type 2 diabetes were largely recruited through the diabetes outpatient clinic of the university hospital Heidelberg. We additionally recruited patients with type 2 diabetes as well

as healthy control participants via newspaper- and online adds. All participants had to be between 40 and 80 years old. Exclusion criteria for people with type 2 diabetes and the healthy control group were medical conditions and medication that are known to influence the physiological stress response as well as conditions that might interfere with adherence to the study protocols. We therefore excluded participants suffering from Cushing's disease, autoimmune diseases, acute feverish infections, type 1 diabetes, severe heart- liver- or kidney disease, participants who reported having suffered from cancer within the last 3 years, participants who suffered from neurological disease such as Parkinson's disease, epilepsy and dementia or severe psychiatric disease such as schizophrenia or bipolar disorder. We excluded participants with regular intake of steroid-based medication or antidepressant medication as well as intake of antihistamines that could not be paused for study participation. We excluded individuals who smoked more than 10 cigarettes a day, drank regularly more than three alcoholic beverages a day or engaged in other forms of drug use. To participate, people with type 2 diabetes had to be diagnosed with type 2 diabetes by a licensed physician. Healthy control participants were required to have no past or current diagnosis of type 2 diabetes.

This study used data of a subsample of the sample described in Buckert et al. ([19] currently under review) and consists of participants with type 2 diabetes who suffer from at least one diabetes-related complication.

## Stress induction

To induce stress, we used the Trier Social Stress Test (TSST [20]). The TSST is a widely used procedure and has been shown to reliably provoke a stress response in a variety of different samples [21]. At the start of the TSST participants receive instructions for a simulated job interview. The interview then takes place in a separate room in front of two "committee members" and a prominently placed camera. Participants are informed that that they will have to give a speech in front of the committee and that the committee members are trained to analyze participant's behavior. They are then given a short preparation period (5 min) during which the committee members watch them closely and take notes. During the entire duration of the TSST (ca. 14 min), the committee members limit social interaction with participants strictly to the TSST protocol and keep a completely neutral facial expression. In the last part of the TSST, the committee members instruct participants to perform a surprise mental arithmetic task (serial subtraction of high numbers). Participants are debriefed after the subsequent resting period of one hour.

## Questionnaires

To measure participant's subjective psychological stress response, we used visual analogue rating scales (VAS). Feelings of tension, as well as the "stressfulness" of the TSST were rated on a continuous scale from 1 to 10. Symptoms of autonomic neuropathy were assessed using the German Version of the survey of autonomic symptoms (SAS) [22]. The Survey consists of 12 items in men and 11 items in women and inquires vasomotor, gastrointestinal, orthostatic, urinary, sudomotor and symptoms as well as erectile dysfunction. Items assess symptom presence as well as symptom severity which is rated on a scale from 1 ("the symptom bothers me . . .not all") to 5 (". . .a lot"). Sum scores are calculated for symptom presence (men: 0–12; women: 0–11) as well as symptom severity (men: 0–60; women: 0–55).

## Procedure

Participants were screened for eligibility via telephone. They were then sent the study information as well as a questionnaire on demographic data via mail. All participants were instructed

to abstain from intense physical activity and alcohol consumption the night before study participation, to get up at least 1.5 hours before their appointment and to postpone intake of medication until after study participation.

All participants arrived on site between 8:30 and 9:30 am. After they provided written informed consent the ECG logger was attached. Subsequently, participants filled in the first VAS. Participants then received instructions for the TSST and were accompanied to a separate room were the TSST took place. Immediately after the stress test, participants filled in the second VAS including their appraisal of the stressfulness of the situation. During the following resting period, participants provided a third rating on the VAS 45 minutes after the TSST, filled in the SAS and provided a urine sample. After completion of the experimental protocol, participants went through a medical examination for the assessment of diabetes-associated complications. A graphical depiction of the study procedures can be found in the S1 Fig.

## Medical examination

Symptoms of peripheral neuropathy were assessed by the Neuropathy Symptom Score and the Neuropathy Disability Score [23]. A diagnosis of peripheral neuropathy was given if patients reached a score of 3 or more on one of these scales. Diagnosis of retinopathy was determined via funduscopy. A diagnosis of nephropathy was given when the albumin-creatinine-ratio (calculated as urinary albumin/(urinary creatinine/100)) of the urine sample was above 30 mg/g.

## ECG sampling and HRV analysis

An ambulatory, 5-lead ECG logger (Schiller Medilog AR12 Plus) was used to perform ECG recordings. This ECG logger has a sampling frequency of 8000 Hz, using oversampling to achieve a high resolution and signal-to-noise ratio. The electrodes were placed on the manubrium, on the right side, close to anterior axillary line, under the left clavicula on the left side on the mid-clavicular line, and on the right sternal border. The ECG recording ran for approximately 1.5 hours. Relevant events (such as the beginning and end of the stress test) were marked in the recording. Using these markers, we extracted six three-minute ECG-samples from the recording for each participant. We used ultra-short samples of three minutes to achieve a detailed assessment of the cardiac autonomic response to the stress test. The baseline sample was recorded while participants were in a seated position ("Baseline") approximately 15 minutes after arrival. The second sample ("anticipation") was recorded during the preparation period of the TSST while participants were already seated in the TSST room. The third ("stress test 1") sample was recorded during participant's speech and the fourth ("stress test 2") during the arithmetic task. Sample five ("post-stress") was recorded directly after the TSST and the last sample ("recovery") 15 minutes after the TSST.

Raw ECG data were processed using Kubios HRV software version 3.3 (Kubios Oy, Kuopio, Finland [24]). For QRS detection, Kubios applies an algorithm based on the Pan-Tompkins algorithm [25] and marks R-waves to calculate RR intervals and create a heart period time series. The marked QRS complexes were inspected visually and edited were necessary. Technical and physiological artefacts (ectopic beats, arithmetic events) were identified visually as well as through a threshold-based correction algorithm using a threshold value of 0.35 sec. The correction algorithm applies median filtering to calculate local average intervals and compares every RR interval value to the respective local average interval value. RR intervals that differ more than 0.35 sec from the locale average are marked as possible artefacts for removal. If samples consisted of more than 5% corrected or removed beats they were excluded from further analysis. Samples were detrended automatically using the smoothness priors method [26].

A parametric autoregressive modeling approach was used to estimate power spectral density [27]. High frequency (HF) and low frequency (LF) HRV were extracted using the established frequency bands (HF: 0.15–0.4 Hz; LF: 0.04–0.15 Hz [28]). While LF HRV is used as an index of both parasympathetic as well as sympathetic cardiac modulatory influence, reflecting for example modulation of vasomotor tone, HF HRV is strongly related to respiratory sinus arrhythmia and thus indexes mainly (but not exclusively) vagal modulation of cardiac activity [29–31].

## Statistical analyses

All statistical analyses were conducted using IBM SPSS Statistics for Windows version 26 [32]. We used *t*-tests to compare participants with type 2 diabetes and healthy controls on VAS ratings. Longitudinal multilevel modelling via SPSS MIXED according to the procedure described in Peugh [33] was used to analyze the effect of type 2 diabetes on the cardiac autonomic stress response. HR and HRV data were log-transformed to fulfill normality assumptions and outliers ($-3 > z > 3$) that remained after transformation were excluded from the analysis. Continuous predictor variables (BMI, age) were grad mean centered.

We modeled individual heart rate and HRV samples (baseline, anticipation, stress test 1, stress test 2, post-stress and recovery) as levels one units while participants were modeled as level two units. In longitudinal multilevel modeling, level one and two can be understood as two regression equations predicting heart rate and HRV. The level one equation contains only time as predictor as all other predictors (type 2 diabetes and control variables) refer to participants rather than individual samples and are consequently modeled as level two predictors within the level two equation. In respect to the curved nature of the data we included time as linear as well as quadratic ($time^2$) effect [34]. In this procedure it is possible to include cross-level interactions in the model. Therefore, not only the differences between people with type 2 diabetes overall (level two) can be determined but also differences in linear or quadratic change in heart rate and HRV over time. We controlled for the influence of age, gender, BMI and hypertensive medication (beta blockers, ACE inhibitors, calcium channel blockers, angiotensin receptor blockers) by including these variables as additional predictors in the models.

For all three outcome variables (heart rate, LF and HF HRV), we specified a random intercept fixed slope model. In respect of the longitudinal nature of the data we employed a first-order autoregressive variance structure. Both models contained the following predictors: time (linear and quadratic), type 2 diabetes and the control variables as well as the cross-level interactions between time and type 2 diabetes and between time and each of the control variables.

## Results

### Sample description

Data of six participants with type 2 diabetes and of 3 healthy control participants had to be excluded from the analysis due to a high ratio of artefacts in the ECG-samples. Our final sample (n = 98) consisted of 51 participants with type 2 diabetes and 47 healthy controls. The mean age of the sample was 64.3 (*SD* = 7.6) and 41.8% were female. Participants with type 2 diabetes had significantly less years of school education and showed a significantly higher mean BMI (participants with type 2 diabetes: 30.4 (*SD* = 5.4), healthy controls 25.7 (*SD* = 3.5)). Please see Table 1 for more details.

We assessed diabetic complications in participants with type 2 diabetes and found 22% suffered from retinopathy, 34% showed albuminuria indicating nephropathy and 90% showed signs of peripheral polyneuropathy. Participants with type 2 diabetes reported on average 2.6

**Table 1. Sample description and differences between type 2 diabetes patients and healthy control participants.** Data are depicted as means (standard deviation) or $n$ (percentage). Group differences were tested using $t$-test for continuous variables as well as $chi^2$- tests for categorical variables.

| | Participants with type 2 diabetes (n = 51) | Healthy controls (n = 47) | $p$ |
|---|---|---|---|
| **Gender** | male: 29(56.9%), female: 22(43.1%) | male: 28(59.6%), female: 19(40.4%) | .786 |
| **Age** (years) | 65.4(7.3) | 63.2(7.9) | .148 |
| **School Education** | | | .018 |
| <10 years of education | 19(37.3%) | 5(10.9%) | |
| 10 years of education | 12(23.5%) | 11(23.9%) | |
| >10 years of education | 19(37.3%) | 29(63.0%) | |
| Does not apply | 1(2.0%) | 1(2.2%) | |
| **Marital Status** | | | .548 |
| Single | 4(7.8%) | 7(14.9%) | |
| Married | 35(68.6%) | 30(63.8%) | |
| Divorced | 6(11.8%) | 7(14.9%) | |
| Widowed | 6(11.8%) | 3(6.4%) | |
| **BMI** | 30.4(5.4) | 25.7(3.5) | < .001 |
| **Illness duration** (years) | 13,3(10.9) | | |
| **Hba1c IFCC** | 56.2(12.2) | 36.1(3.8) | < .001 |
| **Hba1c** | 7.3%(1.1%) | 5.4%(0.4%) | |
| **Medication** | | | |
| Insulin | 18(35.3) | | |
| Other diabetic medication | 40(78,4) | | |
| Beta blockers | 12(23.5) | 3(6.4) | .019 |
| Other hypertensive medication | 32(62.7) | 10(21.3) | < .001 |
| **Diabetic Complications** | | | |
| Retinopathy | 11(21.9) | | |
| Albuminuria | 17(34.0) | | |
| Polyneuropathy | 46(90.2) | | |
| SAS Symptom Score (0–12) | 2.6(2.4) | | |
| SAS Symptom Impact Score (0–60) | 7.5(8.0) | | |

SAS = survey of autonomic symptoms.

other hypertensive medication = ACE inhibitors, calcium channel blockers, angiotensin receptor blockers.

other diabetic medication = metformin, sulfonylureas, GLP-1 receptor agonists, gliptins, gliflozins.

($SD$ = 2.4; range = 0–10) symptoms of autonomic neuropathy according to the SAS and had a mean symptom impact score of 7.5 ($SD$ = 8.0) with a range of 0 to 35.

## Heart rate

Fig 1 depicts the mean heart rate values of participants with type 2 diabetes and healthy control participants throughout the stress test. Longitudinal multilevel modeling showed a significant, linear effect of time ($est.$ = 0.08, $p < .001$) implying an overall increase in heart rate, and a negative quadratic effect of time ($est.$ = -0.02, $p < .001$) indicating an inverse U-shape of HR data over time. Type 2 diabetes had no significant main effect ($est.$ = 0.01, $p = .35$), implying no difference between the groups in mean heart rate. However, the model showed a significant interaction between type 2 diabetes and linear ($est.$ = -0.02, $p = .02$) as well as quadratically modeled time ($est.$ = 0.003, $p = .02$) thus revealing an attenuated HR increase in response to the stress test as well as a significantly slower HR recovery for participants with type 2 diabetes. More detailed information on predictor estimates can be found in Table 2.

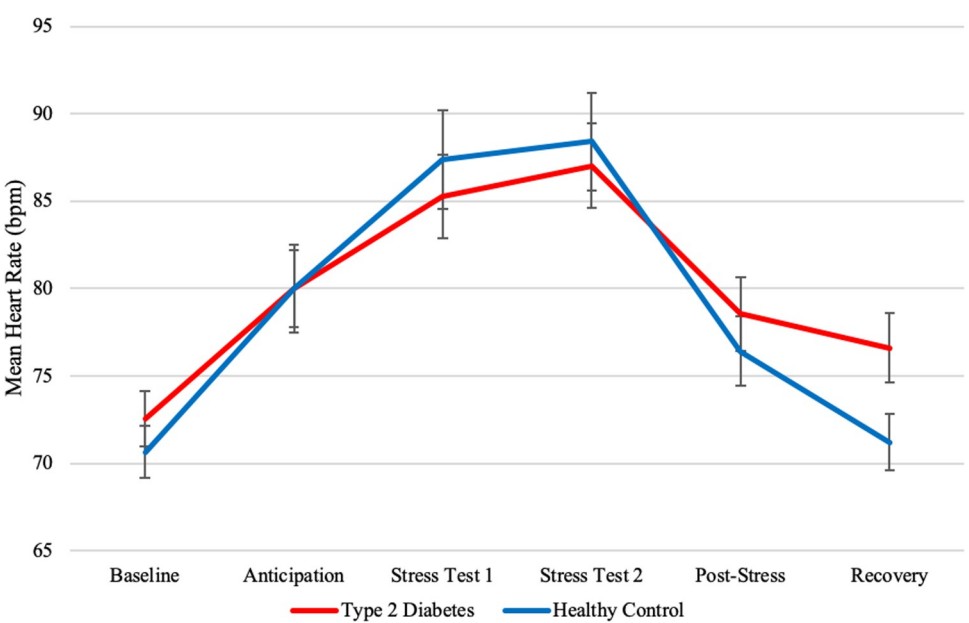

**Fig 1. Mean heart rates and standard errors of type 2 diabetes patients and healthy controls before and after stress induction. Note:** Values depict averages of 3-minute HR-samples. Time from baseline to anticipation was on average 27 minutes. The stress test took on average 14 minutes. Time from post-stress to recovery was approximately 15 minutes.

## Heart rate variability

**HF HRV.** Mean values of HF HRV for participants with type 2 diabetes and healthy controls can be found in Fig 2. The model on HF HRV showed a significant, positive quadratic (*est.* = 0.04, *p* = .03) effect for time, indicating a U-shape of HF HRV over time. Type 2 diabetes had no significant main effect (*est.* = -0.21, *p* = .09), implying no difference between the groups in average HF HRV and no significant interaction with linear (*est.* = -0.03, *p* = .64) or quadratic (*est.* = -0.004, *p* = .77) time, indicating no differences between the groups in change of HF HRV over time. More detailed information on predictor estimates can be found in Table 3.

**LF HRV.** Mean values of LF HRV for participants with type 2 diabetes and healthy controls can be found in Fig 3. The model showed no significant, linear (*est.* = -0.04, *p* = .67) or quadratic (*est.* = 0.02, *p* = .24) effects for time, indicating no change in LF HRV throughout the stress test. Type 2 diabetes had a significant, negative main effect (*est.* = -0.27, *p* = .03) on LF HRV, implying overall lower LF HRV in type 2 diabetes patients. The model showed no significant interaction between type 2 diabetes and linear (*est.* = 0.11, *p* = .08) or quadratic time (*est.*

**Table 2. Multilevel model on log(Heart rate): Estimates of fixed effects.**

| Parameter | *Estimate* | *SE* | *t* | *p* |
|---|---|---|---|---|
| Intercept | 1.84 | 0.02 | 76.18 | < .001 |
| Time (linear) | 0.08 | 0.01 | 8.39 | < .001 |
| Time (quadratic) | -0.02 | 0.002 | -9.04 | < .001 |
| Type 2 Diabetes | 0.01 | 0.02 | 0.35 | .73 |
| Type 2 Diabetes* Time (linear) | -0.02 | 0.01 | -2.33 | .02 |
| Type 2 Diabetes* Time (quadratic) | 0.003 | 0.001 | 2.46 | .02 |

Note: Effects of age, gender, BMI and hypertensive medication were controlled for.

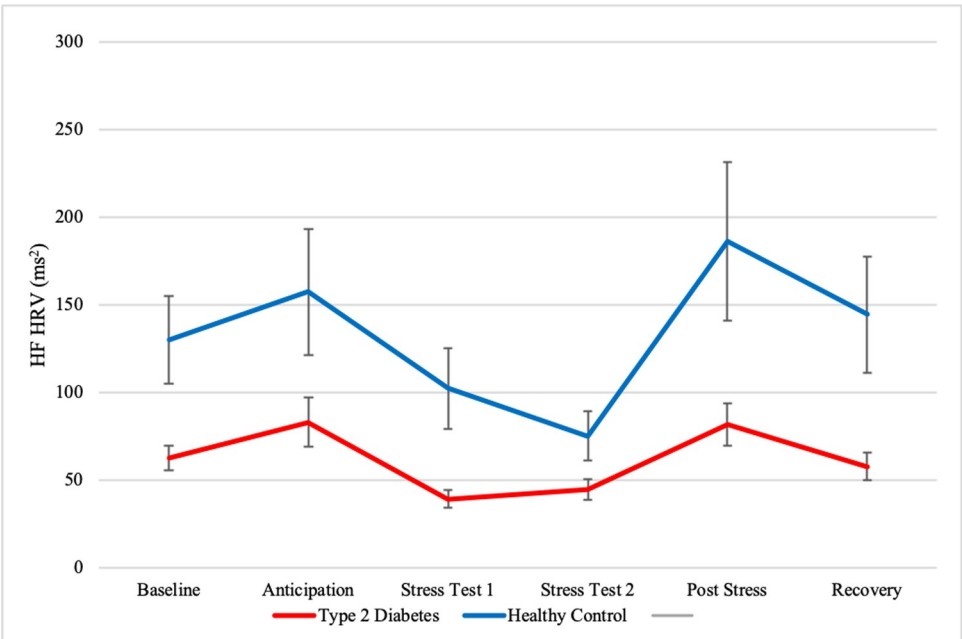

**Fig 2. Mean HF HRV and standard errors of type 2 diabetes patients and healthy controls before and after stress induction. Note:** Values depict averages of 3-minute HRV-samples. Time from baseline to anticipation was on average 27 minutes. The stress test took on average 14 minutes. Time from post-stress to recovery was approximately 15 minutes.

= -0.02, $p$ = .06) indicating no differences between the groups in change in LF HRV over time. More detailed information on predictor estimates can be found in Table 4.

## Psychological stress response

In both groups, reported tension increased significantly ($p < .001$) from an average baseline value of 3.4 ($SD$ = 2.0) to 5.4 ($SD$ = 2.1) after the TSST. Average ratings of tension for both groups are depicted in Fig 4. Independent sample $t$-tests showed no differences in self-reported tension between participants with type 2 diabetes and healthy control participants at baseline ($t_{96}$ = 0.29, $p$ = .78) or directly after the stress test ($t_{92}$ = 0.63, $p$ = .53). 45 minutes after the stress test however, participants with type 2 diabetes reported significantly higher levels of tension ($t_{94}$ = 2.11, $p$ = .04, $d$ = 0.43). The groups did not differ in their appraisal of the "stressfulness" of the TSST ($t_{87}$ = 0.32, $p$ = .75).

**Table 3. Multilevel model on log(HF HRV): Estimates of fixed effects.**

| Parameter | Estimate | SE | t | p |
|---|---|---|---|---|
| Intercept | 1.69 | 0.17 | 10.13 | < .001 |
| Time (linear) | -0.14 | 0.09 | -1.62 | .11 |
| Time (quadratic) | 0.04 | 0.02 | 2.25 | .03 |
| Type 2 Diabetes | -0.21 | 0.12 | -1.71 | .09 |
| Type 2 Diabetes* Time (linear) | 0.03 | 0.06 | 0.47 | .64 |
| Type 2 Diabetes* Time (quadratic) | -0.004 | 0.01 | -0.30 | .77 |

Note: Effects of age, gender, BMI and hypertensive medication were controlled for.

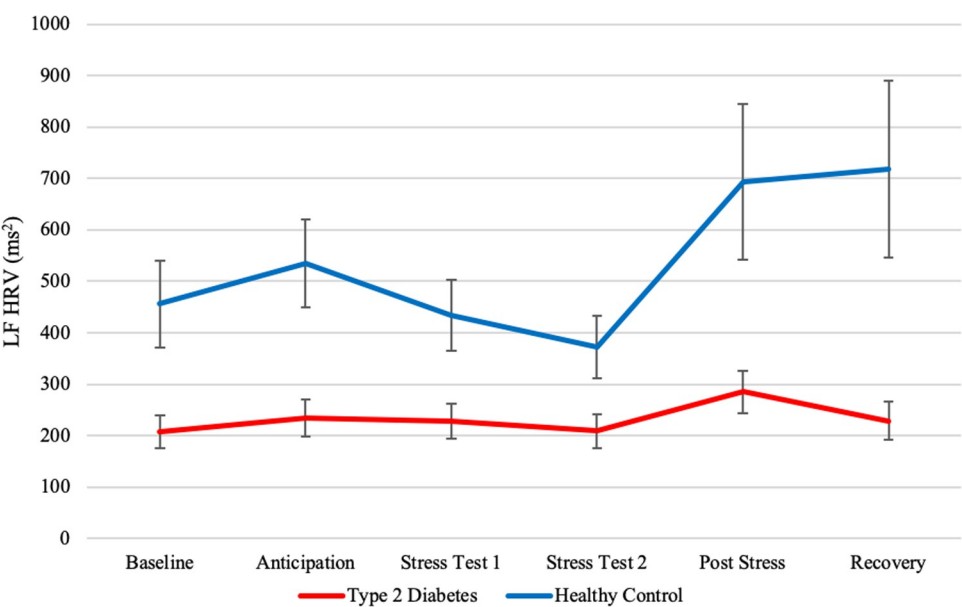

**Fig 3. Mean LF HRV and standard errors of type 2 diabetes patients and healthy controls before, during and after stress induction. Note:** Values depict averages of 3-minute HRV-samples. Time from baseline to anticipation was on average 27 minutes. The stress test took on average 14 minutes. Time from post-stress to recovery was approximately 15 minutes.

## Discussion

The aim of the present study was to examine the cardiac autonomic as well as the psychological stress response of people with type 2 diabetes. We found an anttanuated HR reponse to the stress test and a slower HR recovery accompanied by lower LF HRV in type 2 diabetes patientes but no difference in vagally mediated HF HRV between the groups. The slowed HR recovery was reflected in an increased level of self-reported tension 45 minutes after the stress test.

Participants with type 2 diabetes showed an attenuated heart rate response to psychosocial stress with a attenuated HR reactivity combined with a slowed recovery which was illustrated by the significantly less pronouced curvature of the HR curve in participants with type 2 diabetes. This pattern suggests an imparied ability to respond to environmental demands as well as to downregulate the physiological arousal after stressor cessation. The result is in line with the study by Steptoe et al. [17], who described a comparable heart rate response pattern in a sample with type 2 diabetes. Interestingly, Steptoe et al. [17] excluded type 2 diabetes patients with

**Table 4. Multilevel model on log(LF HRV): Estimates of fixed effects.**

| Parameter | Estimate | SE | t | p |
|---|---|---|---|---|
| Intercept | 2.34 | 0.17 | 14.23 | < .001 |
| Time (linear) | -0.04 | 0.09 | -0.43 | .67 |
| Time (quadratic) | 0.02 | 0.02 | 1.12 | .24 |
| Type 2 Diabetes | -0.27 | 0.12 | -2.20 | .03 |
| Type 2 Diabetes* Time (linear) | 0.11 | 0.06 | 1.79 | .08 |
| Type 2 Diabetes* Time (quadratic) | -0.02 | 0.01 | -1.86 | .06 |

Note: Effects of age, gender, BMI and hypertensive medication were controlled for.

## Psychological Tension

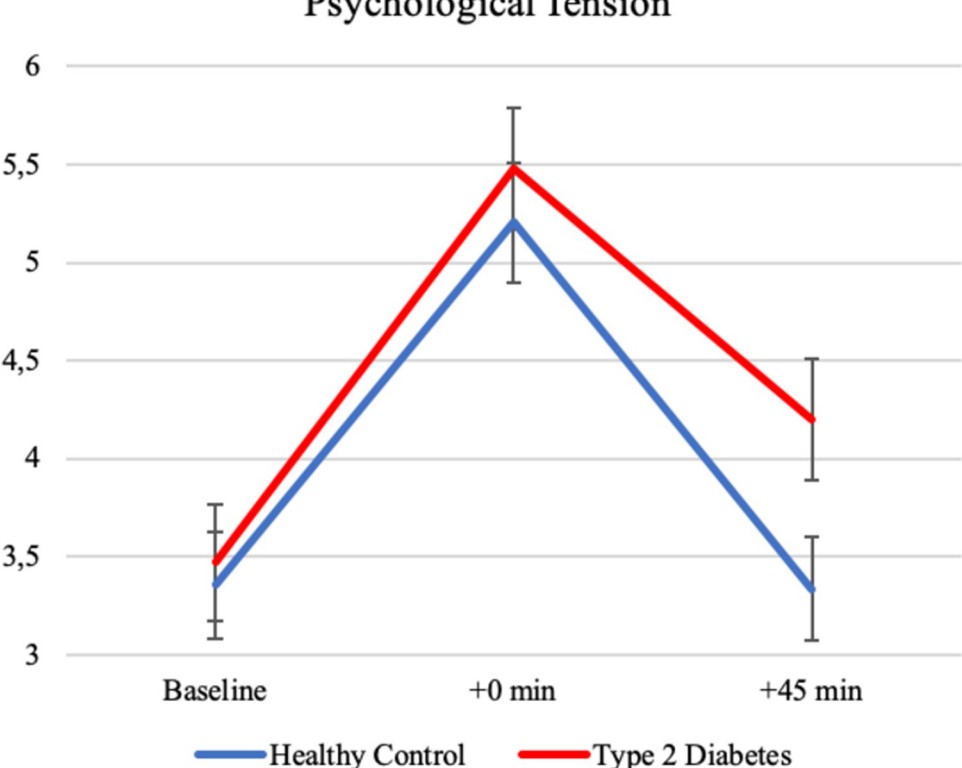

**Fig 4. Mean ratings of subjective tension (0–10) of type 2 diabetes patients and healthy controls before, directly after and 45 minutes after the stress test.**

signs of cardiac autonomic neuropathy. It is thus possible this blunted heart rate response pattern preceeds the symptomatic manifestation of cardiac autonomic neuropathy or may even develop independently.

Surprisingly, we found no group difference in HF HRV, an indicator of vagal modulatory influence, although decreased vagal tone is a common finding in people with type 2 diabetes [4]. Differences in vagal tone between people with type 2 diabetes and healthy controls are commonly assessed under resting conditions and might diminish under stress. LF HRV on the other hand can be understod as an indicator of sympathetic as well as parasympathtic activity. Unlike people with other chronic conditions, people with type 2 diabetes exhibit an overall (sympathetic as well as parasympathetic) decrease in autonomic activity [3]. Our results possibly reflect that relationship and indicate the difference in the cardiac autonomic response to stress in participants with type 2 diabetes may not mainly originate from reduced vagal tone but possibly rather in a decrease in both, parasympathetic and sympathetic activity, illustrated by LF HRV rather than HF HRV.

When considering studies using exercise stress tests, blunted heart rate responses to a stressor are a common findings in samples with type 2 diabetes [35, 36] and are frequently interpreted as a sign of of cardiac autonomic neuropathy [37] but have also been used to predict type 2 diabetes incidence [38]. However, in the context of psychosocial stress, this cardiac response pattern is considered a sign of allostatic load [39] and as thus understood as the result of repeated or chronic activation of the stress system. As chronic stress has been linked to type 2 diabetes [40], decreased autonomic flexibility in people with type 2 diabetes could be

understood as both, a disease-related change in the autonomic nervous system and as a consequence of the wear and tear of chronic stress.

The differences between the groups in the cardiac autonomic stress response were reflected in participants' subjective stress recovery, suggesting the observed changes in the autonomic stress response are possibly concurrent with participants' self-perception. While our study design does not allow assumptions on causality, this finding nevertheless stresses the importance of considering the interplay of physiologal and psychological factors in type 2 diabetes research and treatment and expands the understanding of diabetes-related stress: while suffering from type 2 diabetes is in many ways a stessor initself, changes in the physiological stress response may additionally impair stress regulation in people with type 2 diabetes. Psychologically, a prolonged recovery period is often linked to maladaptive emotion regulation strategies such as self-criticism and rumination [41]. As effective emotion regulation has been shown to be relevant in dealing with diabetes distress [42] as well as glycaemic control [43] treatments may target emotion- and stressregulation to attenuate the effects of stress in people with type 2 diabetes.

There are some limitattions that need to be taken into account when interpreting the results of this study. As our study is largely based on cross-sectional data, it is not possible to determine whether the differences in the cardiac autonomic stress response are mainly a consequnce of type 2 diabetes and its complications or a precedessor of type 2 diabetes [44]. Furthermore, we only assessed short term HRV, to make coprehensive conclusions regarding cardiac autonomic function, longer recordings are necessary. Lastly, although we controlled for the effect of medication, a confounding influence of hypertensive medication cannot be ruled out entirely.

Futur research could advance the understanding of the effect of type 2 diabetes on the cardiac autonomic stress response by comparing participants suffering from type 2 diabetes with and without cardiac autonomic neuropathy. The effects of an altered cardiac autonomic stress response on people's day to day life could be investigated within ambulatory settings and the relationship between an altered cardiac autonomic stress response and common consequences of allostatic load such as depression could be explored [45].

## Conclusions

The impairment of the autonomic nervous system found in people with type 2 diabetes is reflected in the HR response to stress—but not in the HRV response—and partially mirrored in the psychological stress response. Further research on the mechanisms that link stress and type 2 diabetes should be expanded to include the autonomic nervous system. Our results stress the importance of considering the complex interplay of psychosocial stress and disease-related changes in the physiological stress system in type 2 diabetes research. Clinicians should be aware of the psychosomatic aspects of type 2 diabetes and its complications and consider acute and chronic stress as an important factor in the treatment of type 2 diabetes.

## Supporting information

**S1 Fig.**
(PDF)

## Acknowledgments

We would like to thank all the participants for their participation in the study, as well as Thomas Flemming, Carmen Streibel, Nikola Henningsen and Karla Berger for their contribution to the study. For the publication fee we acknowledge financial support by the Deutsche

Forschungsgemeinschaft within the funding programme "Open Access Publikationskosten" as well as by Heidelberg University.

## Author Contributions

**Conceptualization:** Nelly Lou Monzer, Mechthild Hartmann, Magdalena Buckert, Peter Nawroth, Beate Wild.

**Data curation:** Nelly Lou Monzer, Kira Wolff.

**Formal analysis:** Nelly Lou Monzer.

**Funding acquisition:** Mechthild Hartmann, Magdalena Buckert, Peter Nawroth, Hans-Christoph Friederich, Beate Wild.

**Investigation:** Nelly Lou Monzer, Magdalena Buckert, Zoltan Kender.

**Methodology:** Nelly Lou Monzer, Mechthild Hartmann, Magdalena Buckert, Kira Wolff, Stefan Kopf, Beate Wild.

**Project administration:** Nelly Lou Monzer, Mechthild Hartmann, Magdalena Buckert.

**Resources:** Peter Nawroth, Stefan Kopf, Zoltan Kender, Hans-Christoph Friederich.

**Software:** Nelly Lou Monzer.

**Supervision:** Mechthild Hartmann, Magdalena Buckert, Beate Wild.

**Visualization:** Nelly Lou Monzer.

**Writing – original draft:** Nelly Lou Monzer.

**Writing – review & editing:** Mechthild Hartmann, Magdalena Buckert, Kira Wolff, Peter Nawroth, Stefan Kopf, Zoltan Kender, Hans-Christoph Friederich, Beate Wild.

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
