## [Decision Letter · Decision Letter 0]

1 Nov 2021

PONE-D-21-31231The Cardiac Autonomic Response to Acute Psychological Stress in Type 2 DiabetesPLOS ONE

Dear Dr. Monzer,

Thank you for submitting your manuscript to PLOS ONE. After careful consideration, we feel that it has merit but does not fully meet PLOS ONE’s publication criteria as it currently stands. Therefore, we invite you to submit a revised version of the manuscript that addresses the points raised during the review process.

We look forward to receiving your revised manuscript.

Kind regards,

Yih-Kuen Jan, PhD

Academic Editor

PLOS ONE

Journal Requirements:

3. Please ensure that you refer to Figure 3 in your text as, if accepted, production will need this reference to link the reader to the figure.

Reviewers' comments:

Reviewer's Responses to Questions

**Comments to the Author**

1. Is the manuscript technically sound, and do the data support the conclusions?

Reviewer #1: Yes

Reviewer #2: Partly

2. Has the statistical analysis been performed appropriately and rigorously? 

Reviewer #1: Yes

Reviewer #2: N/A

3. Have the authors made all data underlying the findings in their manuscript fully available?

Reviewer #1: Yes

Reviewer #2: Yes

4. Is the manuscript presented in an intelligible fashion and written in standard English?

Reviewer #1: Yes

Reviewer #2: No

5. Review Comments to the Author

Reviewer #1: ABSTRACT

1. Line 22: The abbreviation for HRV should come from "heart rate variability".

2. Line 27-28: Although the SDNN activity of diabetic patients was lower than that of the healthy group, there was no statistical difference in the response of SDNN and RMSSD to the stress test between the two groups. Therefore, the description of "impaired autonomic flexibility pattern" does not seem appropriate.

INTRODUCTION

This part is well written.

MATERIALS & METHODS

1. Line 120-134, page 6-7: It may be easier for the readership if the authors provided a diagram of experimental procedure, so that the reader can more clearly know the time points of the six experimental phases. And confirm the time taken by the participants to complete the entire experiment. Was it 56 minutes (Line 225-227)? Moreover, it can also integrate the point-in-time correlation of various parameters such as HR, HRV, VAS, and TSST.

2. Line 145-146, page7: Generally, the basic unit of HRV analysis, whether it is time domain analysis or frequency domain analysis, is based on a 5-minute period. Why is the 3-minute ECG record used here for analysis? Please explain the reason.

3. Line 156-160, page8: Generally, time-domain analysis is more commonly used to analyze periodic or longer-period records. Frequency domain analysis seems to be more commonly used to observe autonomic changes in a short period of time. It also uses 3-5 minutes as the unit for analysis, which seems to be more in line with the author's design of 6 sampling and 3-minute analysis (Line 1458-146). In addition, frequency domain analysis can additionally observe changes in sympathetic activity, because according to the literature, the sympathetic activity of diabetic patients will also be affected. Based on the above two factors, it is suggested that the author can try to observe the results of this research data by frequency domain analysis.

RESULTS

Overall, this section reads well.

DISCUSSION

1. Line 260-261, page 15: The data in Figures 3 and 4 cannot support the description of “impaired autonomic flexibility”. Because there was no difference in HRV response between the two groups after the TSST test.

2. Line 263, page 15: What does "Cardiac response pattern" refer to? The meaning here is unclear, please specify exactly which physiological parameter response?

3. Line 277-279, page 16: This sentence means that age and diabetes have a bottoming effect on the reduction of RMSSD activity. Is there any evidence in the literature? Isn't their influence cumulative?

4. Line 310-313, page 17: The effects of exercise training and breathing exercise intervention on autonomic flexibility mentioned in this paragraph seem to be irrelevant to this study?

CONCLUSION

The conclusions of the paper were very general and deserve better specificity. The conclusion of the paper should say more about how to use this information to make a change in practical application, and comment on the specific research that is suggested to move the science forward.

Reviewer #2: 1. In the subsection “Participants”, the number of the participants and their demographic data should be clearly described. Also, the exclusion criteria were not well described.

2. For the subsection “ECG Sampling and HRV Analysis”, the major concerns are as follows.

(1) The authors used SDNN and RMSSD of HRV to indicate the “sympathovagal balance” and “cardiac vagal outflow”, respectively. How about the frequency domain indices of HRV?

(2) What types of signals were recorded using the ECG logger? The ECG signals or “Heart rate and heart rate variability samples”?

(3) Why a sampling frequency of 8000 Hz was used?

(4) The positions of five ECG electrodes.

(5) How to perform the procedure “artifact correction”? what does the term “artifact rate” mean? (line 155)

3. The “Results” section was not well organized.

4. There are a lot of incorrect or inappropriate expressions. Only in the abstract, the meanings of many terms such as “Multilevel analyses”, “attenuated HR”, “decreased HR curvature”, “HR recovery”, and “stress system” are ambiguous.

6. PLOS authors have the option to publish the peer review history of their article (what does this mean?). If published, this will include your full peer review and any attached files.

Reviewer #1: No

Reviewer #2: **Yes: **Fuyuan Liao

---

## [Author Response · Author response to Decision Letter 0]

25 Nov 2021

Reviewer #1: ABSTRACT

1. Line 22: The abbreviation for HRV should come from "heart rate variability".

Thank you for pointing out this mistake. We have corrected it. (s. line 22)

2. Line 27-28: Although the SDNN activity of diabetic patients was lower than that of the healthy group, there was no statistical difference in the response of SDNN and RMSSD to the stress test between the two groups. Therefore, the description of "impaired autonomic flexibility pattern" does not seem appropriate.

Thank you. We have replaced the expression in question (s. line 27-28)

INTRODUCTION

This part is well written.

MATERIALS & METHODS

1. Line 120-134, page 6-7: It may be easier for the readership if the authors provided a diagram of experimental procedure, so that the reader can more clearly know the time points of the six experimental phases. And confirm the time taken by the participants to complete the entire experiment. Was it 56 minutes (Line 225-227)? Moreover, it can also integrate the point-in-time correlation of various parameters such as HR, HRV, VAS, and TSST.

Thank you for this suggestion. We have provided a supplementary figure depicting the experimental procedure including point-in-time correlations between the VAS and the physiological variables. As depicted in the figure, the study procedures took approximately 2 hours. 

2. Line 145-146, page7: Generally, the basic unit of HRV analysis, whether it is time domain analysis or frequency domain analysis, is based on a 5-minute period. Why is the 3-minute ECG record used here for analysis? Please explain the reason.

We used 3-min intervals to achieve a more detailed assessment of the stress response during the stress test. The stress test lasted only approximately 14 minutes and we aimed for 3 samples assessing the HR and HRV response to stress anticipation, the speech task and the surprise arithmetic task. Due to participants varying reactions to the stress test, there was a (small) variance in the exact timing of the stress test. Using 3-minute samples ensured three non-overlapping ECG samples for all participants. To ensure comparability, we wanted to keep all ECG-samples the same length and therefore only used 3-minute samples. 

We have inserted a short explanation of the ECG sample length in the Methods section (s. lines 151-152).

As suggested in your third comment below, we have replaced RMSSD and SDNN with HF and LF HRV. For both HF and LF HRV, samples of <5 min. have been shown to be reliable. 

See for example:

Castaldo, R., Montesinos, L., Melillo, P., James, C., & Pecchia, L. (2019). Ultra-short term HRV features as surrogates of short term HRV: a case study on mental stress detection in real life. BMC medical informatics and decision making, 19(1), 1-13.

Baek, H. J., Cho, C. H., Cho, J., & Woo, J. M. (2015). Reliability of ultra-short-term analysis as a surrogate of standard 5-min analysis of heart rate variability. Telemedicine and e-Health, 21(5), 404-414.

Salahuddin, L., Cho, J., Jeong, M. G., & Kim, D. (2007, August). Ultra short term analysis of heart rate variability for monitoring mental stress in mobile settings. In 2007 29th annual international conference of the ieee engineering in medicine and biology society (pp. 4656-4659). IEEE.

3. Line 156-160, page8: Generally, time-domain analysis is more commonly used to analyze periodic or longer-period records. Frequency domain analysis seems to be more commonly used to observe autonomic changes in a short period of time. It also uses 3-5 minutes as the unit for analysis, which seems to be more in line with the author's design of 6 sampling and 3-minute analysis (Line 1458-146). In addition, frequency domain analysis can additionally observe changes in sympathetic activity, because according to the literature, the sympathetic activity of diabetic patients will also be affected. Based on the above two factors, it is suggested that the author can try to observe the results of this research data by frequency domain analysis.

We have replaced RMSSD and SDNN with HF and LF power. RMSSD was highly correlated with HF power and SDNN with LF power and our results did not change when using frequency domain analysis (s. results section). Correlations and results of both frequency and time-domain analysis can be compared below on page 5 and 6 of the document "Response to Reviewers". 

RESULTS

Overall, this section reads well.

DISCUSSION

1. Line 260-261, page 15: The data in Figures 3 and 4 cannot support the description of “impaired autonomic flexibility”. Because there was no difference in HRV response between the two groups after the TSST test.

Thank you. We have replaced the expression in question (s. line 281).

2. Line 263, page 15: What does "Cardiac response pattern" refer to? The meaning here is unclear, please specify exactly which physiological parameter response?

Thank you for pointing this out. We have revised the sentence. (s. line 281)

3. Line 277-279, page 16: This sentence means that age and diabetes have a bottoming effect on the reduction of RMSSD activity. Is there any evidence in the literature? Isn't their influence cumulative?

As we found no conclusive evidence for neither a floor effect, nor a cumulative effect, we revised the section deleted the sentence.

4. Line 310-313, page 17: The effects of exercise training and breathing exercise intervention on autonomic flexibility mentioned in this paragraph seem to be irrelevant to this study?

Thank you. We have deleted the paragraph in question. 

CONCLUSION

The conclusions of the paper were very general and deserve better specificity. The conclusion of the paper should say more about how to use this information to make a change in practical application, and comment on the specific research that is suggested to move the science forward.

Thank you for your suggestion. We expanded the respective paragraphs (s. lines 334-339 and 345-347).

Reviewer #2: 

1. In the subsection “Participants”, the number of the participants and their demographic data should be clearly described. Also, the exclusion criteria were not well described.

Thank you for pointing this out. We have revised the section (s. lines 78-81).

Number of participants and demographic data can be found in the Results section (sample description, s. lines 211-215 and table 1). We have expanded this section as well.

2. For the subsection “ECG Sampling and HRV Analysis”, the major concerns are as follows.

(1) The authors used SDNN and RMSSD of HRV to indicate the “sympathovagal balance” and “cardiac vagal outflow”, respectively. How about the frequency domain indices of HRV?

We have replaced RMSSD and SDNN with HF and LF power. RMSSD was highly correlated with HF power and SDNN with LF power and our results did not change when using frequency domain analysis (s. Results section). Correlation and results of both frequency and time-domain analysis can be compared on page 5 and 6 of the document "Response to Reviewers".

(2) What types of signals were recorded using the ECG logger? The ECG signals or “Heart rate and heart rate variability samples”?

Thank you for pointing out this inaccuracy. We have revised the paragraph in question (s. line 144-145).

(3) Why a sampling frequency of 8000 Hz was used?

The ECG system we used uses oversampling to improve resolution and signal-to-noise ratio. We have included an explanation in the respective section (s. line 145-146).

(4) The positions of five ECG electrodes.

Thank you for your suggestion. We have included a description of the placement of the electrodes in line 146-148.

(5) How to perform the procedure “artifact correction”? what does the term “artifact rate” mean? (line 155)

Thank you for your suggestion. We have expanded this section and included a more detailed description of the procedure (s. lines 160-170).

3. The “Results” section was not well organized.

Thank you for pointing this out. We have reorganized the results section. Results on HR and HRV are now placed before the results on psychological tension.

4. There are a lot of incorrect or inappropriate expressions. Only in the abstract, the meanings of many terms such as “Multilevel analyses”, “attenuated HR”, “decreased HR curvature”, “HR recovery”, and “stress system” are ambiguous.

Thank you for pointing this out. We have tried to be more specific (s. lines 26-29, 34-35, 189, 230, 279, 281)

---

## [Decision Letter · Decision Letter 1]

13 Jan 2022

PONE-D-21-31231R1The Cardiac Autonomic Response to Acute Psychological Stress in Type 2 DiabetesPLOS ONE

Dear Dr. Monzer,

Thank you for submitting your manuscript to PLOS ONE. After careful consideration, we feel that it has merit but does not fully meet PLOS ONE’s publication criteria as it currently stands. Therefore, we invite you to submit a revised version of the manuscript that addresses the points raised during the review process.

We look forward to receiving your revised manuscript.

Kind regards,

Yih-Kuen Jan, PhD

Academic Editor

PLOS ONE

Journal Requirements:

Reviewers' comments:

Reviewer's Responses to Questions

**Comments to the Author**

1. If the authors have adequately addressed your comments raised in a previous round of review and you feel that this manuscript is now acceptable for publication, you may indicate that here to bypass the “Comments to the Author” section, enter your conflict of interest statement in the “Confidential to Editor” section, and submit your "Accept" recommendation.

Reviewer #1: (No Response)

2. Is the manuscript technically sound, and do the data support the conclusions?

Reviewer #1: Partly

3. Has the statistical analysis been performed appropriately and rigorously? 

Reviewer #1: Yes

4. Have the authors made all data underlying the findings in their manuscript fully available?

Reviewer #1: Yes

5. Is the manuscript presented in an intelligible fashion and written in standard English?

Reviewer #1: Yes

6. Review Comments to the Author

Reviewer #1: Thanks for your reply to my questions. Your revisions were generally accepted. There are still some minor issues that require your reply. The questions are as follows:

LINE 28 & 279: The sentence described in the text (Line 28 & 279) appears to be misleading: "This pattern was accompanied by significantly decreased low frequency HRV ..... between the groups."

The term "decreased" to describe the LF HRV response in the text appears to be misleading. Because there was no difference in LF HRV in group-by-time interaction in the stress response, only the LF HRV in the DM group was consistently lower than that in the healthy group. Therefore, it is suggested that the word "lower" is more appropriate to replace "decrease" here.

LINE 32-33 & 341-342: "The impairment of the autonomic nervous system commonly found in people with type 2 diabetes likely affects the stress response." This sentence in the conclusion is too excessive and vague. In fact, the difference in ANS was only in the LF, and the stress response only affected the heart rate and subjective perception, however, there was no significant difference in the ANS response. Therefore, it is recommended to make more specific and precise conclusions to avoid misunderstandings.

7. PLOS authors have the option to publish the peer review history of their article (what does this mean?). If published, this will include your full peer review and any attached files.

Reviewer #1: No

---

## [Author Response · Author response to Decision Letter 1]

11 Feb 2022

LINE 28 & 279: The sentence described in the text (Line 28 & 279) appears to be misleading: "This pattern was accompanied by significantly decreased low frequency HRV ..... between the groups."

The term "decreased" to describe the LF HRV response in the text appears to be misleading. Because there was no difference in LF HRV in group-by-time interaction in the stress response, only the LF HRV in the DM group was consistently lower than that in the healthy group. Therefore, it is suggested that the word "lower" is more appropriate to replace "decrease" here.

Thank you for your suggestion! We have replaced the word “decreased” with the word “lower” in Line 28 and 279.

LINE 32-33 & 341-342: "The impairment of the autonomic nervous system commonly found in people with type 2 diabetes likely affects the stress response." This sentence in the conclusion is too excessive and vague. In fact, the difference in ANS was only in the LF, and the stress response only affected the heart rate and subjective perception, however, there was no significant difference in the ANS response. Therefore, it is recommended to make more specific and precise conclusions to avoid misunderstandings.

Thank you! We have replaced the sentence in question with a more precise conclusion. See lines 32-34 & 342-344

---

## [Decision Letter · Decision Letter 2]

28 Feb 2022

The Cardiac Autonomic Response to Acute Psychological Stress in Type 2 Diabetes

PONE-D-21-31231R2

Dear Dr. Monzer,

We’re pleased to inform you that your manuscript has been judged scientifically suitable for publication and will be formally accepted for publication once it meets all outstanding technical requirements.

Kind regards,

Yih-Kuen Jan, PhD

Academic Editor

PLOS ONE

Additional Editor Comments (optional):

Reviewers' comments:

Reviewer's Responses to Questions

**Comments to the Author**

1. If the authors have adequately addressed your comments raised in a previous round of review and you feel that this manuscript is now acceptable for publication, you may indicate that here to bypass the “Comments to the Author” section, enter your conflict of interest statement in the “Confidential to Editor” section, and submit your "Accept" recommendation.

Reviewer #1: All comments have been addressed

2. Is the manuscript technically sound, and do the data support the conclusions?

Reviewer #1: Yes

3. Has the statistical analysis been performed appropriately and rigorously? 

Reviewer #1: Yes

4. Have the authors made all data underlying the findings in their manuscript fully available?

Reviewer #1: Yes

5. Is the manuscript presented in an intelligible fashion and written in standard English?

Reviewer #1: Yes

6. Review Comments to the Author

Reviewer #1: Thanks to the author's revision, this paper appears to have been improved. This article is now suitable for acceptance and publication.

7. PLOS authors have the option to publish the peer review history of their article (what does this mean?). If published, this will include your full peer review and any attached files.

Reviewer #1: No

---

## [Editor Report · Acceptance letter]

10 Mar 2022

PONE-D-21-31231R2 

The Cardiac Autonomic Response to Acute Psychological Stress in Type 2 Diabetes 

Dear Dr. Monzer:

I'm pleased to inform you that your manuscript has been deemed suitable for publication in PLOS ONE. Congratulations! Your manuscript is now with our production department. 

Kind regards, 

on behalf of

Dr. Yih-Kuen Jan 

Academic Editor

PLOS ONE